# The impact of rehabilitation frequency on the risk of stroke in patients with rheumatoid arthritis

Yuan-Yang Cheng[1,2], Shin-Tsu Chang[1,3,4,5], Chung-Lan Kao[2,6], Yi-Huei Chen[7], Ching-Heng Lin[7]*

1 Department of Physical Medicine and Rehabilitation, Taichung Veterans General Hospital, Taichung, Taiwan, 2 School of Medicine, National Yang-Ming University, Taipei, Taiwan, 3 Department of Physical Medicine and Rehabilitation, Tri-Service General Hospital, Taipei, Taiwan, 4 School of Medicine, National Defense Medical Center, Taipei, Taiwan, 5 School of Medicine, Chung Shan Medical University, Taichung, Taiwan, 6 Department of Physical Medicine and Rehabilitation, Taipei Veterans General Hospital, Taipei, Taiwan, 7 Department of Medical Research, Taichung Veterans General Hospital, Taichung, Taiwan

* epid@vghtc.gov.tw

**Data Availability Statement:** The data underlying this study is from the National Health Insurance Research Database (NHIRD), which has been transferred to the Health and Welfare Data Science Center (HWDC). Interested researchers can obtain

## Abstract

### Background

Patients with rheumatoid arthritis are at higher risk of developing stroke due to augmented systemic inflammation. While regular exercise has been shown to reduce inflammation in human subjects, the purpose of our study is to determine whether increased frequency of rehabilitation is protective against stroke or not.

### Methods

A total of 16,224 rheumatoid arthritis patients with a catastrophic illness certificate were identified in our database from 2000 to 2006, and 663 of them developed stroke before the end of 2013. After statistically matching for age, sex, and the time interval between the diagnosis of rheumatoid arthritis and stroke, 642 patients without stroke were identified as the control group. Analyses with Student's t test and Chi-square test were done to compare the clinical characteristics of patients with and without stroke, and logistic regression analysis was then performed to evaluate the odds ratio of stroke.

### Results

Higher odds ratio of stroke was revealed in patients with hypertension, diabetes mellitus, and moderate degree of rheumatoid arthritis disease severity. Furthermore, more than 40 rehabilitation sessions per year reduced the risk of developing stroke in patients with moderate disease severity.

### Conclusions

Physical activities brought by more rehabilitation sessions are beneficial and should be encouraged in patients with rheumatoid arthritis, particularly for those with higher disease activity but not taking biologic agents.

the data through formal application to the HWDC, Department of Statistics, Ministry of Health and Welfare, Taiwan (http://dep.mohw.gov.tw/DOS/np-2497-113.html).

**Funding:** CHL received grants from Taichung Veterans General Hospital, Taiwan (TCVGH-1087322D). The funders had no role in study design, data collection and analysis, decision to publish, or preparation of the manuscript.

**Competing interests:** The authors have declared that no competing interests exist.

# Introduction

Rheumatoid arthritis (RA) is an autoimmune disease characterized by systemic inflammation. [1] Although the synovial joints are the main target of autoantibodies in patients with RA, many other organs or tissues may also be involved, such as skin, eyes, lungs, kidneys, blood vessels, and salivary glands.[2] Because of the involvement of systemic inflammation, many diseases closely related to inflammation may also be induced by RA. Patients with RA often suffer from decreased quality of life and disability due to destruction of joints in later life. It is thus vital to conduct thorough, ongoing evaluations in RA patients in order to identify and avoid comorbid diseases that may further exacerbate disability.

Stroke is caused by the interruption of blood supply to the brain. Both ischemic and hemorrhagic stroke can have a considerable negative impact on patients' quality of life due to symptoms such as hemiplegia, hemianopia, aphasia, and dysphagia. Well-known risk factors of stroke include hypertension, type 2 diabetes mellitus (DM), dyslipidemia, physical inactivity, obesity, and cigarette smoking, etc..[3] In addition, risk of stroke has also been demonstrated to be higher in patients with RA.[4] As endothelial and systemic inflammation has long been considered to play a central role in the pathogenesis of stroke,[5] efforts should be directed toward reducing systemic inflammation in patients with RA to lower the incidence of stroke. The current mainstream treatment approach centers on medications, which range from corticosteroids and disease-modifying anti-rheumatic drugs to biological agents.[6] Additionally, exercise and physical activities also appear to play an important role in reducing systemic inflammation.[7] Exercise is proven to be beneficial for patients with RA in reducing fatigue, [8] lowering daily activity limitations, and improving both oxygen uptake and muscle strength.[9] However, to date, no studies have been conducted to clarify the relationship between the exercise frequency and risk of stroke development in patients with RA. Because rehabilitation programs for RA patients usually include strengthening exercises for the major muscles of the four limbs as well as cardiopulmonary endurance training, they constitute an important component of daily exercise in RA patients. We designed a retrospective nested case-control study to elucidate the association between rehabilitation frequency and stroke incidence in RA patients. We hypothesized that RA patients receiving more rehabilitation training sessions would have a lower incidence of stroke.

# Materials and methods

## The database

This study used data from Taiwan's National Health Insurance Research Database (NHIRD), which was administered by the National Health Insurance (NHI) Administration, the single largest medical health insurance institution in Taiwan. More than 99% of Taiwan's approximately 23 million residents, including foreign nationals, are enrolled in Taiwan's NHI program.[10] Therefore, data from the NHIRD reflect the general medical health status of the entire population of Taiwan. The data that are made available to researchers include encrypted patients' data such as date of birth, sex, medical diagnoses in the form of ICD-9-CM (International classification of diseases, ninth revision, clinical modification) codes, date of hospital admission and discharge, the procedures received, and the prescribed medications that are covered by the NHI. The database has been used extensively to conduct medical epidemiology studies in Taiwan.[11] Hence, it provides a valuable resource to investigate the relationship between the number of rehabilitation programs received and the incidence of stroke in patients with RA.

## The study samples

The study employed a nested case-control design, which was approved by the Institutional Review Board of Taichung Veterans General Hospital in Taiwan. (No. CE13152) Patients with an unambiguous diagnosis of RA in Taiwan are certified with a catastrophic-illness card (CIC) by the NHI Administration, and it allows these patients to be exempted from most medical costs related to RA when they visit health facilities. In order to avoid mistaken diagnosis in the database, only patients with a CIC for RA were recruited in our study, and the period of recruitment was from January 1st, 2000 to December 31st, 2006. Moreover, the diagnosis of stroke was established only if an ICD-9 code from 430 to 438 was used for the major diagnosis at hospital admission after radiological confirmation. After excluding patients with a diagnosis of stroke before RA or age younger than 18 years old, a total of 16,244 patients with RA were enrolled as the study subjects. Among them, 663 patients (4.08%) developed stroke before December 31st, 2013.

After matching for age, sex, and the time interval between RA and stroke diagnosis, we identified an additional 642 RA patients without stroke occurrence before the end of 2013. An index date was assigned to each of these patients corresponding to the date of stroke attack in the study group. After excluding patients with a time interval between the date of RA diagnosis and stroke or index date less than one year, there were 591 cases of RA with subsequent stroke and 582 cases of RA without stroke. The flowchart of the study participants selection is summarized in Fig 1.

## The comorbidity definitions, groups setting, and study design

Medical comorbidities including hypertension, DM, and hyperlipidemia were taken into consideration in our study. Each patient was defined as having a particular medical comorbidity if it was registered as the major diagnosis at least three times in a patient's outpatient records and at least one time within one year before the index or stroke date.

In order to study the effect of more rehabilitation sessions on patients with RA, we defined patients receiving 40 rehabilitation training sessions per year as the cutoff point. Rehabilitation programs for RA patients in Taiwan usually involve educating patients how to perform strengthening exercises for the four limbs to stabilize the vulnerable joints, cardiopulmonary endurance training, and splints fitting, as well as providing suggestions related to activities of daily living. As suggestions from American College of Sports Medicine, strengthening exercises are usually designed as 60–80% of 1 repetition maximum, 8–12 repetitions for 2–4 sets, 2–3 days per week, while cardiopulmonary endurance training emphasizes on exercises that maintaining the heart rate around 40–60% of heart rate reserve for an accumulation of 150 minutes per week.[12] Forty rehabilitation sessions per year roughly approximates to rehabilitation visits once per week under the instruction of professional physical therapists after subtracting national holidays and vacations.

In order to study the effect of rehabilitation in RA patients with different disease severity, we divided the RA patients into three groups. The advantage of our database is that the dosage of medications was available. Since medications prescriptions in Taiwan were regulated by the rules set by Taiwan's NHI Administration, certain criteria of disease severity should be fulfilled in order to be eligible for treatment with biologic agents. Therefore, we can use the medication types and dosage to classify the severity of diseases in studies utilizing Taiwan's NHIRD. We defined patients in the severe group as those who had ever received biologic agents, including etanercept, adalimumab, golimumab, abatacept, rituximab, and tocilizumab within one year before the stroke or index date. Patients in the moderate group were defined as those who had ever received methotrexate >420mg or prednisolone >280mg in one outpatient visit. The

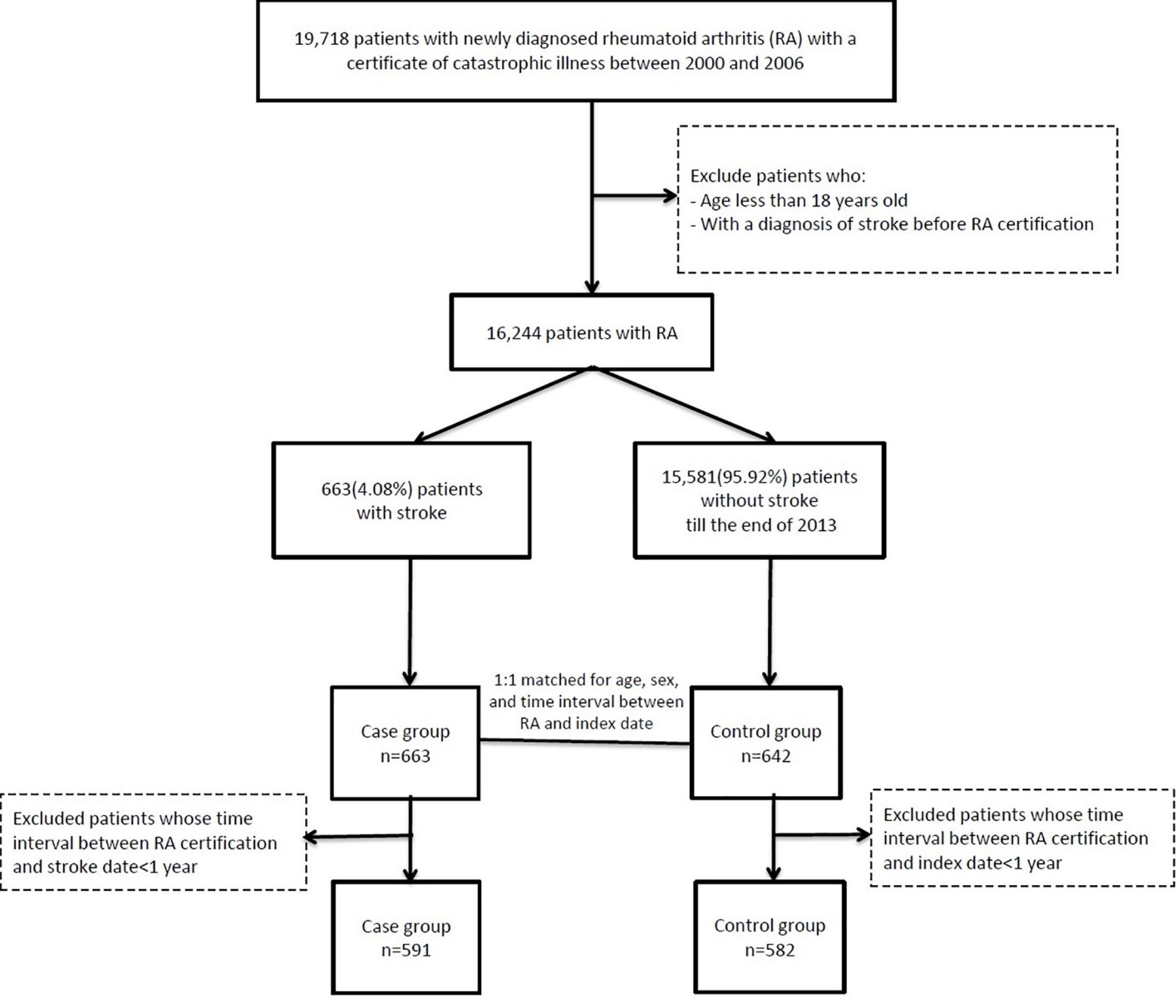

**Fig 1. The flowchart of our study.** The enrollment of the study cases is illustrated in this figure.

remaining patients were assigned to the mild group. The effects of different rehabilitation frequencies were determined in these three groups of RA disease severity.

## Statistical analysis

We used SAS software, version 9.4 (SAS Institute, Cary, NC, USA) for all statistical analyses in this study. Student's t test and Chi-square test were done to determine the statistical differences in age, sex, medical comorbidities, including hypertension, DM, and hyperlipidemia, number of rehabilitation sessions per year, and RA disease severity between the patients with stroke and without stroke. Next, multivariable logistic regression analysis was conducted to determine the odds ratio of stroke based on age, sex, medical comorbidities, number of

**Table 1. Clinical characteristics of RA patients with and without stroke.**

| Variables | Total (n = 1173) | | Without Stroke (n = 582) | | With Stroke (n = 591) | | P-value |
|---|---|---|---|---|---|---|---|
| | n | (%) | n | (%) | n | (%) | |
| Age, years (mean±SD) | 62.1±10.7 | | 62.0±10.6 | | 62.1±10.7 | | 0.938[a] |
| Gender | | | | | | | 0.680 |
| female | 834 | (71.1) | 417 | (71.6) | 417 | (70.6) | |
| male | 339 | (28.9) | 165 | (28.4) | 174 | (29.4) | |
| Hypertension | | | | | | | <0.001 |
| No | 632 | (53.9) | 369 | (63.4) | 263 | (44.5) | |
| Yes | 541 | (46.1) | 213 | (36.6) | 328 | (55.5) | |
| DM | | | | | | | <0.001 |
| No | 958 | (81.7) | 510 | (87.6) | 448 | (75.8) | |
| Yes | 215 | (18.3) | 72 | (12.4) | 143 | (24.2) | |
| Hyperlipidemia | | | | | | | 0.002 |
| No | 993 | (84.7) | 512 | (88.0) | 481 | (81.4) | |
| Yes | 180 | (15.3) | 70 | (12.0) | 110 | (18.6) | |
| Number of rehabilitation sessions per year | 12.9±58.0 | | 14.1±69.7 | | 11.7±43.6 | | 0.474[a] |
| 0–39 | 1078 | (91.9) | 528 | (90.7) | 550 | (93.1) | 0.142 |
| ≧40 | 95 | (8.1) | 54 | (9.3) | 41 | (6.9) | |
| Severity of RA | | | | | | | 0.083 |
| Mild | 794 | (67.7) | 402 | (69.1) | 392 | (66.3) | |
| Moderate | 259 | (22.1) | 114 | (19.6) | 145 | (24.5) | |
| Severe | 120 | (10.2) | 66 | (11.3) | 54 | (9.1) | |
| Time interval between RA and index date, years (mean±SD) | 6.0±3.0 | | 6.0±3.0 | | 6.0±3.0 | | 0.875[a] |

[a] T test, Chi-squared test or Fisher's Exact Test for all other p-values.

rehabilitation sessions, and RA disease severity with statistical adjustment for all the variables mentioned above. In order to clarify the impact of different rehabilitation frequencies in the three groups of RA disease severity, a stratified analysis according to the disease severity was done with statistical adjustment of age, sex, hypertension, DM and hyperlipidemia. A p value less than 0.05 was considered significant in our study.

## Results

The clinical characteristics of RA patients with and without stroke are shown in Table 1. Because the two groups of patients were matched for age, sex, and time interval between the diagnosis of RA and index date, no statistical differences in these three parameters were noted between patients with and without stroke. The average time interval between the diagnosis of RA and stroke attack was 6.0±3.0 years in our study. Significantly greater prevalence rates of hypertension, DM, and hyperlipidemia were noted in the stroke patients. However, the number of rehabilitation sessions per year and the disease severity of RA did not differ significantly between these two groups of patients, and the p values were 0.474 and 0.083, respectively. After grouping the patients according to the cutoff number of rehabilitation sessions, the Chi-square test still did not achieve statistical significance (p = 0.083).

To study the effect of the aforementioned parameters on stroke attack, logistic regression analysis was done with statistical adjustment for all the parameters including age, sex, disease comorbidities, number of rehabilitation sessions, and severity of RA. The odds ratios of stroke in RA patients with hypertension or DM were 2.10 and 1.92, respectively, with

**Table 2. Adjusted odds ratios of stroke for age, sex, hypertension, DM, hyperlipidemia, number of rehabilitation per year, and severity of RA in the multivariable logistic regression analysis.**

| Variable | Adjusted Odds ratio | 95% CI | P-value |
|---|---|---|---|
| Age, years | 0.99 | (0.98–1.00) | 0.213 |
| Sex | | | |
| female | 1.00 | — | — |
| male | 1.14 | (0.87–1.48) | 0.339 |
| Hypertension | | | |
| No | 1.00 | — | — |
| Yes | 2.10 | (1.64–2.69) | <0.001* |
| DM | | | |
| No | 1.00 | — | — |
| Yes | 1.92 | (1.38–2.66) | <0.001* |
| Hyperlipidemia | | | |
| No | 1.00 | — | — |
| Yes | 1.29 | (0.92–1.83) | 0.144 |
| Number of rehabilitation sessions per year | | | |
| 0–39 | 1.00 | — | — |
| ≧40 | 0.66 | (0.42–1.02) | 0.061 |
| Severity of RA | | | |
| Mild | 1.00 | — | — |
| Moderate | 1.37 | (1.02–1.84) | 0.034* |
| Severe | 0.82 | (0.55–1.23) | 0.339 |

*p<0.05

p values <0.001. However, RA patients with hyperlipidemia did not have a higher risk of developing stroke. (odds ratio = 1.29, p = 0.144) Regarding disease severity of RA, only patients with moderate severity had a higher risk of stroke compared to the group with mild severity (odds ratio = 1.37, p = 0.034). Furthermore, RA patients receiving more rehabilitation sessions did not have a significant reduction in risk of stroke attack (odds ratio = 0.66, p = 0.061). The results are shown in Table 2. Since physical activities brought by more frequent rehabilitation visits had been considered to be beneficial for stroke prevention, the result above seems to be unreasonable. Therefore, further stratified analysis on the effect of rehabilitation frequencies was done according to the disease severity. More rehabilitation sessions only resulted in a risk reduction effect in patients with moderate disease severity, (odds ratio = 0.32, p = 0.039) as shown in Table 3.

## Discussion

This is the first study to analyze the relationship between rehabilitation frequencies and incidence of stroke in RA patients. We discovered that only patients with moderate RA severity had a higher incidence of stroke. Moreover, it was the only group in which a higher frequency of rehabilitation was shown to have protective effect against stroke.

The method of classifying RA disease severity in administrative databases has been investigated in several studies.[13,14] One such study that has been widely discussed in the literature proposes a claims-based index for RA severity (CIRAS),[15] which includes tests for rheumatoid factor, inflammatory markers, number of chemistry panels and platelet counts ordered, number of rehabilitation and rheumatology visits, and Felty's syndrome. Although this model was demonstrated to have a moderate correlation with a medical records-based index of RA

**Table 3. Adjusted odds ratio of stroke associated with number of rehabilitation sessions per year, stratified by disease severity of RA in the multivariable logistic regression analysis.**

| Variables | Adjusted Odds ratio | 95% CI | P-value |
|---|---|---|---|
| **Severity of RA:Mild** | | | |
| Number of rehabilitation sessions per year | | | |
| 0–39 | 1.00 | — | — |
| ≧40 | 0.75 | (0.44–1.26) | 0.272 |
| **Severity of RA:Moderate** | | | |
| Number of rehabilitation sessions per year | | | |
| 0–39 | 1.00 | — | — |
| ≧40 | 0.32 | (0.11–0.95) | 0.039* |
| **Severity of RA:Severe** | | | |
| Number of rehabilitation sessions per year | | | |
| 0–39 | 1.00 | — | — |
| ≧40 | 0.97 | (0.26–3.63) | 0.961 |

*p<0.05

severity,[16] many of the items in this index cannot be obtained in the database we used, such as blood tests including rheumatoid factor, inflammatory markers, chemistry panels and platelet counts. Another study attempted to determine disease severity by assessing lifetime exposure to various medications including corticosteroid, disease-modifying antirheumatic drugs (DMARD) and biologic agents.[17] However, this method could not distinguish between RA patients in the highest and lowest quartiles of disease severity. Nevertheless, the advantage of our database is that the dosage of medications was available, in contrast to the aforementioned study. Therefore, we were able to classify the patients' disease severity by the dosage of medications they received. In addition, according to well-defined rules stipulated by Taiwan's National Health Insurance program, patients are required to fulfill certain criteria of disease severity in order to be eligible for treatment with biologic agents, such as a Disease Activity Score (DAS 28)[18] higher than 5.1 and treatment failure for six months using at least two kinds of DMARDs confirmed by photographic evidence and X-ray study. Hence, we classified the patients that had ever received biologic agents as the most severe group, and the other groups were also defined according to the dosage of methotrexate and prednisolone the patients had ever received as shown on their outpatient records. This was found to be an objective way to classify RA disease severity in previous studies that used NHIRD [19,20].

Several epidemiological studies demonstrated a significantly higher risk of stroke in RA patients, and the odds ratio ranged from 1.18 (95% confidence interval 1.09–1.28)[21] to 2.98 (95% confidence interval 1.89–4.70).[22] Data from a meta-analysis revealed an odds ratio of 1.64 (95% confidence interval 1.32–2.05) for ischemic stroke and 1.68 (95% confidence interval 1.11–2.53) for hemorrhagic stroke in RA patients.[23] The risk of recurrent stroke was also higher in RA patients.[24] Because the relationship between RA and cerebrovascular diseases has been thoroughly evaluated in previous investigations, this topic was not the primary focus of the present study. Instead, we compared the stroke risk among different levels of RA disease severity. Our results showed a significantly higher odds ratio of stroke in the moderate severity group compared to the mild group, but not the severe group. By definition, biologic agents were only prescribed in the most severe group of RA patients in our study. Therefore, a possible reason is the anti-inflammatory effect of the biologic agents which would have reduced the likelihood of developing stroke in the most severe group in our study. For instance, etanercept was shown to protect the rat brain against ischemic stroke by inhibiting tumor necrosis factor

(TNF) alpha and downregulating microglial activation.[25] Another biologic agent, adalimumab, could reduce endothelial dysfunction and arterial stiffness, and thus ameliorate carotid atherosclerosis.[26] Rituximab, another frequently prescribed biologic agent, was also revealed to have similar effects on endothelial function.[27] Hence, patients in the severe group, who were thought to have the highest risk of stroke, did not have a significantly higher risk compared to the mild group. On the contrary, without the potent anti-inflammatory effect of biologic agents, patients in the moderate group had a higher risk of developing stroke.

Exercise and physical activity can reduce inflammation and cardiovascular risk in RA patients.[28] In addition to inflammation, a sedentary lifestyle in RA patients also contributes to a significantly elevated cardiovascular risk, which can also be reverted by regular physical activity.[29] Our study revealed that a rehabilitation frequency of more than 40 sessions per year could reduce the risk of stroke in RA patients with moderate disease severity. This group of patients had a somewhat high degree of disease severity, but had never been protected by the potent biologic agents before their stroke attack. This is the circumstance in which exercise could exert its anti-inflammatory effect to protect these patients from stroke attack. Since rehabilitation doctors and physical therapists in Taiwan always design exercise programs for patients according to ACSM's Guidelines for Exercise Testing and Prescription,[12] the comprehensive rehabilitation programs in Taiwan regularly constitute of strengthening and aerobic parts of exercise training. The protective effect of exercise could also be observed in other groups in our study results. The odds ratio of stroke was 0.75 in the mild group, and 0.97 in the severe group. However, no statistical significance could be established, and the mechanism underlying these phenomena warrant further research.

This study is the first of its kind to evaluate the effect of rehabilitation frequency on the incidence of stroke among different levels of RA disease severity. The major strength of our study is the use of a population-based administrative database with a nested case-control design, which allowed us to track the results of a large sample of RA patients without the concern of immortal time bias.[30] However, there were at least five limitations in our study. First, the number of rehabilitation sessions might not have accurately reflected the amount of physical activity in the daily life of RA patients. Nevertheless, in this group of patients, a willingness to receive rehabilitation sessions almost every week for one year indicated a far greater motivation to exercise compared with those who did not attend weekly exercise sessions. These RA patients received instructions in a tailored exercise program from physical therapists every week, and thus it is reasonable to expect that they would maintain their exercise habits. Second, certain risk factors for stroke such as obesity, smoking, alcohol consumption, and dietary habits were not available in our database, and were therefore not possible to control or adjust for them in the analysis. However, these stroke risk factors are also risk factors for hypertension,[31] DM[32] and hyperlipidemia.[33] By statistically adjusting for these medical comorbidities, it was possible at least in part, to control for these risk factors simultaneously. Third, while medical comorbidities including hypertension, diabetes and hyperlipidemia are important contributing factors of stroke, only statistical adjustments were done instead of matching. The reason lies in that if these medical comorbidities in our study were all matched, the case number in the control group will be very few, rendering subsequent statistical analysis impossible and meaningless. Therefore, only age, sex, and the time interval between RA and stroke diagnosis were statistically matched between the two study groups. Fourth, the disease severity of RA patients was determined by the medications prescribed rather than the use of a validated disease severity index such as DAS28.[18] As certain items in the DAS28, such as the number of joints with tenderness or swelling, the erythrocyte sedimentation rate, and the patient global health status, are not available in NHIRD, it was not appropriate to define the disease severity groups using this index in our study. However, DAS28 is one of the criteria for initiation of

biologic agent treatment in National Health Insurance regulations, and doctors always adjust the medication dosages according to the disease severity they observe in clinical practice. Thus, the use of medications prescribed as a surrogate to define disease severity in our study was reasonable. Finally, the type, intensity, frequency and duration of exercise incorporated in the rehabilitation sessions may influence the incidence of stroke. However, the information of detailed exercise programs mentioned above cannot be obtained in the database we used. Therefore, further prospective study to address this gap is needed in the future.

## Conclusions

Our study revealed a higher risk of stroke in patients with hypertension, DM, and moderate disease severity in RA patients. Furthermore, more than 40 rehabilitation sessions per year reduced the risk of developing stroke in RA patients with moderate disease severity. Therefore, physical activities brought by more rehabilitation sessions were shown to be beneficial and should be encouraged in RA patients with high disease severity without the protection of biologic agents.

## Supporting information

**S1 STROBE checklist. STROBE statement—Checklist of items that should be included in reports of *case-control studies.***
(DOC)

## Acknowledgments

The authors would like to thank the Healthcare Service Research Center (HSRC) of Taichung Veterans General Hospital for statistical support, and Mr. Peter Wilds for English editing.

## Author Contributions

**Conceptualization:** Yuan-Yang Cheng, Ching-Heng Lin.

**Data curation:** Shin-Tsu Chang, Yi-Huei Chen.

**Formal analysis:** Chung-Lan Kao, Yi-Huei Chen.

**Investigation:** Shin-Tsu Chang, Chung-Lan Kao, Yi-Huei Chen.

**Methodology:** Chung-Lan Kao.

**Software:** Yi-Huei Chen.

**Supervision:** Ching-Heng Lin.

**Validation:** Shin-Tsu Chang, Chung-Lan Kao.

**Writing – original draft:** Yuan-Yang Cheng.

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
