## [Decision Letter · Decision Letter 0]

3 Oct 2019

PONE-D-19-22440

The Impact of Rehabilitation Frequency on the Risk of Stroke in Patients with Rheumatoid Arthritis

PLOS ONE

Dear Mr. Lin,

Thank you for submitting your manuscript to PLOS ONE. After careful consideration, we feel that it has merit but does not fully meet PLOS ONE’s publication criteria as it currently stands. Therefore, we invite you to submit a revised version of the manuscript that addresses the points raised during the review process.

We would appreciate receiving your revised manuscript by Nov 17 2019 11:59PM. To enhance the reproducibility of your results, we recommend that if applicable you deposit your laboratory protocols in protocols.io, where a protocol can be assigned its own identifier (DOI) such that it can be cited independently in the future. For instructions see: http://journals.plos.org/plosone/s/submission-guidelines#loc-laboratory-protocols

We look forward to receiving your revised manuscript.

Kind regards,

Seana Gall

Academic Editor

PLOS ONE

**Journal Requirements:**

We note that you have included the phrase “data not shown” in your manuscript. Unfortunately, this does not meet our data sharing requirements. PLOS does not permit references to inaccessible data. We require that authors provide all relevant data within the paper, Supporting Information files, or in an acceptable, public repository. Please add a citation to support this phrase or upload the data that corresponds with these findings to a stable repository (such as Figshare or Dryad) and provide and URLs, DOIs, or accession numbers that may be used to access these data. Or, if the data are not a core part of the research being presented in your study, we ask that you remove the phrase that refers to these data.

2. Please include your ethics statement and ethics approval details in the Methods section of your manuscript.

**Comments to the Author**

1. Is the manuscript technically sound, and do the data support the conclusions?

Reviewer #1: Partly

Reviewer #2: Partly

2. Has the statistical analysis been performed appropriately and rigorously? 

Reviewer #1: I Don't Know

Reviewer #2: I Don't Know

3. Have the authors made all data underlying the findings in their manuscript fully available?

Reviewer #1: No

Reviewer #2: Yes

4. Is the manuscript presented in an intelligible fashion and written in standard English?

Reviewer #1: No

Reviewer #2: Yes

5. Review Comments to the Author

Reviewer #1: General comments:

In this case-control study, the authors use registry data aiming to investigate whether in people with rheumatoid arthritis, rehabilitation frequency is associated with lesser risk of stroke. People had higher odds of stroke if they had hypertension or diabetes and a greater number of rehabilitation sessions did not reduce the odds of stroke. Stratification by RA severity appears to reduce risk for people with moderate RA participating in more than 40 sessions of rehabilitation per year.

Exercise is an important modifiable risk factor for stroke, and as such the study investigates the effect of exercise on stroke risk in a population with an underlying chronic condition. While this knowledge is important for the RA population, the manuscript could be reviewed to make a more convincing argument for the importance of this study and the study findings. At present, the language overstates the findings and could be toned down.

Specific comments

Materials and methods:

1. An important addition would be greater detail about the rehabilitation sessions. While this may be difficult to ascertain from the registry, the rehabilitation sessions may or may not have included the type and intensity and frequency of exercise needed to plausibly influence the odds of stroke.

2. It would be helpful to include some further justification about the definition of RA severity used for stratification based on medication types.

3. The statistically adjustment information currently in the design section (lines 114-115) would be better placed in the statistical analysis paragraph.

Results:

1. In table 1, the groups are not matched for participant numbers (n=582 v n=591) can the reason for this be clarified in the text and any missing data explained.

2. Additionally, in table 1 – could the authors be more specific about the number of rehabilitation sessions. Is this the total number of sessions over the 6 years of data or a mean number of sessions with SDs? The numbers listed appear very low compared to the dose of exercise that would be required to significantly influence stroke via mechanisms such as modifying BP or diabetes status.

3. Table 2: It would be helpful for the results to reworded to clearly specify what variables are in the final adjusted model. This may be an English language issue, but it is difficult to interpret if the model includes all the variables listed from the table title and the text or if the table is presenting the variables listed. This is important for the reader to help interpret the results of the study and is not clear as written.

Discussion:

1. The discussion is quite long and would benefit from revision to more concisely discuss the study findings. It’s is difficult to read in its present form.

2. The section in the discussion that justifies the method of classifying RA severity appears contradictory. I appreciate that the registry has limited data available – However, previous literature does not fully support the method used in this study. Are there further justifications for the choice of methods used in this study?

3. The finding that the moderately severe RA group were at higher risk of stroke is interesting (lines 220-230) and perhaps a more robust finding than the effect of number of rehabilitation sessions. This could be discussed and highlighted further.

4. Paragraph commencing Line 237: the authors discuss the role of exercise and physical activity and a sedentary lifestyle. However, because of the limited detail about the rehabilitation sessions we do not know what the ‘active ingredient’ of the rehabilitation program is and this significantly reduces the strength of the findings. Perhaps the choice of words would be used consistently as to whether the rehabilitation is cardiovascular exercise, or more general physical activity (unplanned, not about fitness) or sedentary time. All confer different biological effects on cardiovascular and cardiometabolic health. The discussion here is mixed and therefore is less specific than it could be.

5. The authors consistently refer to >40 rehabilitation sessions as being ‘aggressive’. This language should be toned down through the manuscript as it is not in fact a high intensity of intervention.

Reviewer #2: This is an interesting paper looking at the association between rehabilitation frequency and incidence of stroke among people with rheumatoid arthritis.

Some comments for the authors are as below:

Major comments

Design

- I have a big concern regarding the study design since the control group was only matched with age, sex, time interval between RA diagnosis and stroke. The study aim was to investigate the association between rehab frequency and incidence of stroke among people with RA, it is assumed that the control group should ideally have similar age, gender, and health (e.g. comorbidities) as the case group. However, as shown in Table 2, those with stroke had higher rates of hypertension, diabetes and hyperlipidemia compared to those without stroke. Adjusting for these factors may not address the issue.

- It is unclear how investigators dealt with several disadvantages of the study design including retrospective nature, appropriateness of control group and possibility of recall bias.

- Please provide reference(s) for the cut-off point of 40 training sessions for intensive rehab.

Methods & findings

- In the legends of Table 2 and Table 3, Hazard ratios should be changed to Odds ratio since authors performed logistic regression.

- Please clarify the exact logistic regression models used for the findings shown in Table 2 and Table 3. Were they bivariable or multivariable models? It seems that after accounting for age, sex, comorbidities, and RA severity, there was no evidence of a statistically significant association between rehab frequency and stroke incidence.

- Please provide the reason for stratifying analyses by RA severity? Explain by checking interaction.

Discussion

- Some key discussions about the classification of RA severity could be moved up to the design section to support the reasons for using the dosage of medications to categorise the severity (instead of using other methods).

- In the earlier section, the authors stated that this study aim was to investigate the association between rehab frequency and incidence of stroke. However, in paragraph 3 of the discussion part, a different aim was mentioned “Instead, we compared the stroke risk among different levels of RA disease severity”. Please clarify.

- According to my concern above, the interpretation of findings shown in Tables 2 and 3 may need to be changed, and the relevant changes in the discussion part may be required.

- Is there any other potential factor that may influence the association between rehab frequency and incidence of stroke such as type of rehabilitation and the length of treatment? Please also discuss.

Minor comments

- The use of different terms such as aggressive rehabilitation and intensive rehabilitation throughout the paper may be unclear to non-specialists. Please provide definitions.

- STROBE Statement—Checklist of items that should be included in reports of case-control studies

6. PLOS authors have the option to publish the peer review history of their article (what does this mean?). If published, this will include your full peer review and any attached files.

Reviewer #1: No

Reviewer #2: Yes: Hoang T Phan

---

## [Author Response · Author response to Decision Letter 0]

11 Oct 2019

Journal Requirements:

1. The database we analyzed in this study was authorized by Taiwan National Health Insurance Administration, and could only be assessed in the Research Laboratory of our hospital. After statistical analyzing this database, the tables and figure in our manuscript are the only data we’re able to take out from the Research Laboratory. Therefore, I’m afraid that the original whole authorized database owned by our hospital cannot be provided under this circumstance.

2. Due to the anonymous nature of the national database, informed consent cannot be signed by each study subject. However, our study was still approved by the Institutional Review Board of Taichung Veterans General Hospital in Taiwan. (No. CE13152), and the ethics approval statement was included in “The Study Samples” section of “Material and Methods” in our manuscript.

Response to Reviewer #1:

Thanks for the detailed evaluation and critical comments on our study. We agree with your opinion that exercise is an important component to prevent stroke attack, especially in those with higher risk of stroke, such RA patients. We will try to reinforce this important point of view in our manuscript without overstating the presented findings. Thank you very much!

Materials and Methods:

1. We agree that the detailed content of exercise programs designed for RA patients should be mentioned in our manuscript. Since rehabilitation doctors and physical therapists in Taiwan always design exercise programs for patients according to ACSM's Guidelines for Exercise Testing and Prescription by American College of Sports Medicine, we added the following sentences in the section of “The Comorbidity Definitions, Groups Setting, and Study Design”: “As suggestions from American College of Sports Medicine, strengthening exercises are usually designed as 60-80% of 1 repetition maximum, 8-12 repetitions for 2-4 sets, 2-3 days per week, while cardiopulmonary endurance training emphasizes on exercises that maintaining the heart rate around 40-60% of heart rate reserve for an accumulation of 150 minutes per week.” We understand the concern that the rehabilitation sessions shown in our study seemed not enough to influence the risk of stroke. However, it’s just the times of rehabilitation visits recorded in the registry. During each visits, rehabilitation doctors and physical therapists in Taiwan serve as roles of reminding patients to maintain exercise trainings in their daily lives according to the suggestions from ACSM guidelines in addition to just doing exercise trainings. 

2. As mentioned in our discussion, medications prescriptions in Taiwan were regulated by the rules set by Taiwan’s National Health Insurance Administration. Doctors cannot prescribe medications at their own will because National Health Insurance Administration, the single largest health insurance institution in Taiwan, will not pay for the drugs if rules are not followed. Certain criteria of disease severity should be fulfilled in order to be eligible for treatment with biologic agents, such as a Disease Activity Score (DAS 28) higher than 5.1 and treatment failure for six months using at least two kinds of DMARDs confirmed by photographic evidence and X-ray study. Therefore, we can use the medication types and dosage to classify the severity of diseases in studies utilizing the Taiwan NHI research database. Furthermore, the methods were also adopted in the past to stratify RA severity.[1] 

3. We agree that the statistical adjustment information should be placed in the statistical analysis section in our manuscript. Thank you very much for reminding us!

Results:

1. In our study, we identified 663 RA patients with stroke at first, and then we tried to find the control subjects exactly matching for age, sex, and the time interval between RA and stroke diagnosis from 15,581 RA patients without stroke. However, only 642 patients could be matched in our database. After excluding patients whose time interval between RA certification and stroke index date less than 1 year, we obtained 591 RA patients with stroke and 582 RA patients without stroke in our study. There was no missing data. The numbers are not matched just because of we could not find enough patients exactly matching for age, sex and the time interval in our database. 

2. Thanks for your reminding that we should be more specific here. The number of rehabilitation sessions here means the mean number of rehabilitation visits per year from the diagnosis of RA to the onset of stroke or index date. As we mentioned before, it’s just the times of rehabilitation visits recorded in the registry. During each visits, rehabilitation doctors and physical therapists in Taiwan serve as roles of reminding patients to maintain exercise trainings in their daily lives according to the suggestions from ACSM guidelines in addition to just doing exercise trainings.

3. As described in the title of table 2, the variables included in our final adjusted model are age, sex, hypertension, DM, hyperlipidemia, number of rehabilitation per year, and severity of RA. The model is like an equation “y=ax1+bx2+cx3+dx4+ex5+fx6+gx7+h”. While y is stroke, x1 to x7 are age, sex, hypertension, DM, hyperlipidemia, number of rehabilitation per year, and severity of RA, respectively. In the “Statistical Analysis” section, we described “…logistic regression analysis was conducted to determine the odds ratio of stroke based on age, sex, medical comorbidities, number of rehabilitation sessions per year, and RA disease severity with statistical adjustment for all the variables mentioned above”, which specified the variables in our final adjusted model. 

Discussion:

1. We agree that the paragraph of discussion is long in its current form. However, the first paragraph summarizes the study findings, the second one discussed the method stratifying RA disease severity, the third one focused on why patients in the moderate disease severity had a higher risk of stroke, the fourth one showed the relationship between hyperlipidemia and stroke in RA patients in a past study, the fifth one discussed why exercise has protective effect in the moderate disease severity against stroke in our study, and the last one discussed the limitations of our study. After deliberate consideration, we decided to delete the four section of our discussion because the relationship between hyperlipidemia and stroke in RA patients is not the main focus of our study.

2. One of the major limitations of NHI research database is lack of lab data. As we mentioned in the discussion, many components of claims-based index for RA severity (CIRAS) such as rheumatoid factor, inflammatory markers, chemistry panels and platelet counts are not available in our registry. Instead, the advantage of our database is that the dosage of medications was available. Since medications prescriptions in Taiwan were regulated by the rules set by Taiwan’s National Health Insurance Administration, certain criteria of disease severity should be fulfilled in order to be eligible for treatment with biologic agents, such as a Disease Activity Score (DAS 28) higher than 5.1 and treatment failure for six months using at least two kinds of DMARDs confirmed by photographic evidence and X-ray study. Therefore, we can use the medication types and dosage to classify the severity of diseases in studies utilizing the Taiwan NHI research database. Furthermore, the methods were also adopted in the past to stratify RA severity.[1]

3. Thanks for noticing the interesting finding in our study, and we’ve adjusted the sentences of discussion here as: “By definition, biologic agents were only prescribed in the most severe group of RA patients in our study. Therefore, a possible reason is the anti-inflammatory effect of the biologic agents which would have reduced the likelihood of developing stroke in the most severe group in our study.…Hence, patients in the severe group, who were thought to have the highest risk of stroke, did not have a significantly higher risk compared to the mild group. On the contrary, without the potent anti-inflammatory effect of biologic agents, patients in the moderate group had a higher risk of developing stroke.”

4. Thanks for the valuable suggestion, and we’ve added the sentences below to better describe the detail of rehabilitation programs in Taiwan: “Since rehabilitation doctors and physical therapists in Taiwan always design exercise programs for patients according to ACSM's Guidelines for Exercise Testing and Prescription, the comprehensive rehabilitation programs in Taiwan regularly constitute of strengthening and aerobic parts of exercise training.”

5. We agree that more than 40 rehabilitation sessions per year cannot be considered as aggressive. Therefore, the adjectives “aggressive” or “intensive” have been avoided throughout our manuscript, and descriptions with more rehabilitation sessions are used instead. 

References: 

1. Tang KT, Chen YH, Lin CH, Chen DY (2016) Methotrexate is not associated with increased liver cirrhosis in a population-based cohort of rheumatoid arthritis patients with chronic hepatitis C. Sci Rep 6: 33104.

Response to Reviewer #2:

Design:

1. Thanks for the critical comment, and we also agree that risk factors of stroke such as age, sex, and medical comorbidities should all be matched between the study and control groups. However, in real world it’s essentially not possible to match these risk factors between stroke patients and non-stroke patients. In our study, we identified 663 RA patients with stroke at first, and then we tried to find the control subjects exactly matching for age, sex, and the time interval between RA and stroke diagnosis from 15,581 RA patients without stroke. However, only 642 patients could be matched in our database. If medical comorbidities including hypertension, diabetes and hyperlipidemia were to be matched in our study, the case number in the control group will be very few, rendering subsequent statistical analysis impossible and meaningless.

2. One of the major limitations of a retrospective study is the immortal time bias. However, our study avoided this bias by the nested case-control design matching the time interval between the study and control group. Another important issue is recall bias, which is also not possible to exist in our study because of the medical record registry-based in nature. About the appropriateness of the control group, we tried to match the potential confounding factors of stroke including age, sex, and the time interval between RA diagnosis and stroke. Other risk factors such as medical comorbidities were then statistically adjusted in further logistic regression models. We tried to ameliorate the disadvantage of a retrospective study by efforts mentioned above. 

3. After our meticulous searching on the internet, we can still not find the reference for the cut-off times of rehabilitation sessions to be referred as intensive rehabilitation. As mentioned in our manuscript, 40 times of rehabilitation sessions per year roughly approximates to rehabilitation visits once per week after subtracting national holidays and vacations. Thank you for the critical comment, and we admit that the adjectives “intensive” or “aggressive” in our manuscript are not appropriate. Therefore, the adjectives “aggressive” or “intensive” have been avoided throughout our manuscript, and descriptions with more rehabilitation sessions are used instead.

Methods and results:

1. We’ve changed the term “hazard ratio” to “odds ratio”. Thank you very much for reminding us the important naming error.

2. Our logistic regression model is multivariate in nature. The model is like an equation “y=ax1+bx2+cx3+dx4+ex5+fx6+gx7+h”. While y is stroke, x1 to x7 are age, sex, hypertension, DM, hyperlipidemia, number of rehabilitation per year, and severity of RA, respectively. Indeed, after statistical adjustment for age, sex, medical comorbidities, number of rehabilitation sessions and severity of RA, the odds ratio of rehabilitation sessions for stroke attack did not achieve statistical significance. Only in the stratified analysis did we find significant odds ratio in the moderate severity group of RA patients.

3. The reason for us to do stratified analysis by RA severity is because we cannot find significant association between rehabilitation frequency and stroke incidence in the entire cohort we located. Since more physical activities brought by more frequent rehabilitation visits had been considered to be beneficial for stroke prevention, our study result using the entire cohort seems to be unreasonable. Therefore, stratified analysis was needed to clarify the results in the subgroups patients with RA. We also examined the interaction between number of rehabilitation sessions and severity of RA by adding their product terms into the logistic regression model (set at two-tailed p ≤0.05), which showed no significant interaction effects (P = 0.703).

Discussion:

1. We agree that some key discussions about the stratification of RA severity should be moved to the design section for better delineating the reason why we used the dosage of medications to categorize the severity. Therefore, the following paragraphs have been added in the design section: “The advantage of our database is that the dosage of medications was available. Since medications prescriptions in Taiwan were regulated by the rules set by Taiwan’s NHI Administration, certain criteria of disease severity should be fulfilled in order to be eligible for treatment with biologic agents. Therefore, we can use the medication types and dosage to classify the severity of diseases in studies utilizing Taiwan’s NHIRD.”

2. The third section of discussion focused on why patients in the moderate disease severity had a higher risk of stroke. Because the relationship between RA and stroke has been thoroughly studied in the past, comparing the stroke risk among different level of RA disease severity serves as a minor aim of our study. The major aim of this study is still the association between times of rehabilitation sessions and stroke incidence. Relevant changes have been made in our discussions.

3. We agree that different types, intensity and durations of exercise incorporated in the rehabilitation sessions may influence the incidence of stroke. However, the information of detailed exercise programs mentioned above cannot be obtained in the database we used. Nevertheless, rehabilitation doctors and physical therapists in Taiwan always design exercise programs for patients according to ACSM's Guidelines for Exercise Testing and Prescription, and the comprehensive rehabilitation programs in Taiwan regularly constitute of strengthening and aerobic parts of exercise training. We’ve added the above sentences in the discussion paragraph, and relevant details from American College of Sports Medicine were added in the design section. 

Minor comments:

1. We admit that the adjectives “intensive” or “aggressive” in our manuscript are not appropriate. Therefore, the adjectives “aggressive” or “intensive” have been avoided throughout our manuscript, and descriptions with more rehabilitation sessions are used instead.

2. We will provide the STROBE checklist as a supplemental file. Thank you very much for your valuable suggestions.

---

## [Decision Letter · Decision Letter 1]

11 Nov 2019

PONE-D-19-22440R1

The Impact of Rehabilitation Frequency on the Risk of Stroke in Patients with Rheumatoid Arthritis

PLOS ONE

Dear Mr. Lin,

Thank you for submitting your manuscript to PLOS ONE. After careful consideration, we feel that it has merit but does not fully meet PLOS ONE’s publication criteria as it currently stands. Therefore, we invite you to submit a revised version of the manuscript that addresses the points raised during the review process.

In the response to reviewers document please provide (1) the page and line numbers for the changes that have been made and (2) a summary of the actual changes made to the text. Note that reviewer 2 has requested additional changes following the first revision.

We would appreciate receiving your revised manuscript by Dec 26 2019 11:59PM. To enhance the reproducibility of your results, we recommend that if applicable you deposit your laboratory protocols in protocols.io, where a protocol can be assigned its own identifier (DOI) such that it can be cited independently in the future. For instructions see: http://journals.plos.org/plosone/s/submission-guidelines#loc-laboratory-protocols

We look forward to receiving your revised manuscript.

Kind regards,

Seana Gall

Academic Editor

PLOS ONE

Reviewers' comments:

Reviewer's Responses to Questions

**Comments to the Author**

1. If the authors have adequately addressed your comments raised in a previous round of review and you feel that this manuscript is now acceptable for publication, you may indicate that here to bypass the “Comments to the Author” section, enter your conflict of interest statement in the “Confidential to Editor” section, and submit your "Accept" recommendation.

Reviewer #2: (No Response)

2. Is the manuscript technically sound, and do the data support the conclusions?

Reviewer #2: Yes

3. Has the statistical analysis been performed appropriately and rigorously? 

Reviewer #2: Yes

4. Have the authors made all data underlying the findings in their manuscript fully available?

Reviewer #2: No

5. Is the manuscript presented in an intelligible fashion and written in standard English?

Reviewer #2: Yes

6. Review Comments to the Author

Reviewer #2: Design:

1. The authors explained that “If medical comorbidities including hypertension, diabetes and hyperlipidemia were to be matched in our study, the case number in the control group will be very few, rendering subsequent statistical analysis impossible and meaningless.”

Response: Please acknowledge this study limitation in the discussion explaining why two groups were matched only for age, sex, and the time interval between RA and stroke diagnosis but not for comorbidities/other factors (e.g. lower statistical power).

Methods and results:

2. The authors explained that “Our logistic regression model is multivariate in nature. The model is like an equation “y=ax1+bx2+cx3+dx4+ex5+fx6+gx7+h”. While y is stroke, x1 to x7 are age, sex, hypertension, DM, hyperlipidemia, number of rehabilitation per year, and severity of RA, respectively.”

Response: Multivariable models should be mentioned in Tables 2 and 3 (in table legends or using footnotes)

3. “The reason for us to do stratified analysis by RA severity is because we cannot find significant association between rehabilitation frequency and stroke incidence in the entire cohort we located. Since more physical activities brought by more frequent rehabilitation visits had been considered to be beneficial for stroke prevention, our study result using the entire cohort seems to be unreasonable. Therefore, stratified analysis was needed to clarify the results in the subgroups patients with RA.”

Response: So the stratified analysis is more likely to be an ad-hoc analysis which you had not thought about upfront. This was because significant association between rehabilitation frequency and stroke incidence in the entire cohort was observed. To facilitate the readers, I would suggest acknowledge the main findings and clarify the reason for undertaking the ad-hoc analysis and its results in the results section

Discussion

3. “We agree that different types, intensity and durations of exercise incorporated in the rehabilitation sessions may influence the incidence of stroke. However, the information of detailed exercise programs mentioned above cannot be obtained in the database we used. Nevertheless, rehabilitation doctors and physical therapists in Taiwan always design exercise programs for patients according to ACSM's Guidelines for Exercise Testing and Prescription, and the comprehensive rehabilitation programs in Taiwan regularly constitute of strengthening and aerobic parts of exercise training”

Response: Please also acknowledge this limitation in the discussion section. Further research to address this gap may inform the current guidelines.

7. PLOS authors have the option to publish the peer review history of their article (what does this mean?). If published, this will include your full peer review and any attached files.

Reviewer #2: Yes: Hoang Phan

---

## [Author Response · Author response to Decision Letter 1]

17 Nov 2019

1. Thanks for the critical comment that it should be an important limitation in our study. We thus added the following sentences in our paragraph of study limitations in the discussion (Page 16 Line 284): “Third, while medical comorbidities including hypertension, diabetes and hyperlipidemia are important contributing factors of stroke, only statistical adjustments were done instead of matching. The reason lies in that if these medical comorbidities in our study were all matched, the case number in the control group will be very few, rendering subsequent statistical analysis impossible and meaningless. Therefore, only age, sex, and the time interval between RA and stroke diagnosis were statistically matched between the two study groups.”

2. We added “in the multivariate logistic regression analysis” in the table legends of Table 2 (Page 10 Line 189) and Table 3 (Page 11 Line 192). Thanks for your kindly reminding.

3. We revised our manuscript in the result section (Page 10 Line 178) as following: “…RA patients receiving more rehabilitation sessions did not have a significant reduction in risk of stroke attack (odds ratio=0.66, p=0.061). The results are shown in Table 2. Since physical activities brought by more frequent rehabilitation visits had been considered to be beneficial for stroke prevention, the result above seems to be unreasonable. Therefore, further stratified analysis on the effect of rehabilitation frequencies was done according to the disease severity.” Thanks for your valuable suggestion!

4. Thanks for your valuable advice, and we admit that it’s also one important limitation of our study. Therefore, the following sentences were added in our study limitations of discussion (Page 16 Line 299): “…Finally, the type, intensity, frequency and duration of exercise incorporated in the rehabilitation sessions may influence the incidence of stroke. However, the information of detailed exercise programs mentioned above cannot be obtained in the database we used. Therefore, further prospective study to address this gap is needed in the future.” Thanks again for your detailed review and critical comments on our study!

---

## [Editor Report · Decision Letter 2]

5 Dec 2019

PONE-D-19-22440R2

The Impact of Rehabilitation Frequency on the Risk of Stroke in Patients with Rheumatoid Arthritis

PLOS ONE

Dear Mr. Lin,

Thank you for submitting your manuscript to PLOS ONE. After careful consideration, we feel that it has merit but does not fully meet PLOS ONE’s publication criteria as it currently stands. Therefore, we invite you to submit a revised version of the manuscript that addresses the points raised during the review process.

Please see below for additional revision.

We would appreciate receiving your revised manuscript by Jan 19 2020 11:59PM. To enhance the reproducibility of your results, we recommend that if applicable you deposit your laboratory protocols in protocols.io, where a protocol can be assigned its own identifier (DOI) such that it can be cited independently in the future. For instructions see: http://journals.plos.org/plosone/s/submission-guidelines#loc-laboratory-protocols

We look forward to receiving your revised manuscript.

Kind regards,

Seana Gall

Academic Editor

PLOS ONE

Journal Requirements:

Additional Editor Comments:

Thank you for your revisions. Please amend the term 'multivariate' to 'multivariable'. The latter is the correct statistical term for this type of analysis.
---

## [Author Response · Author response to Decision Letter 2]

13 Dec 2019

We’ve amended the term “multivariate” to “multivariable” in our manuscript. Thank you for the critical suggestion!

---

## [Editor Report · Decision Letter 3]

20 Dec 2019

The Impact of Rehabilitation Frequency on the Risk of Stroke in Patients with Rheumatoid Arthritis

PONE-D-19-22440R3

Dear Dr. Lin,

We are pleased to inform you that your manuscript has been judged scientifically suitable for publication and will be formally accepted for publication once it complies with all outstanding technical requirements.

With kind regards,

Seana Gall

Academic Editor

PLOS ONE
---

## [Editor Report · Acceptance letter]

30 Dec 2019

PONE-D-19-22440R3 

The Impact of Rehabilitation Frequency on the Risk of Stroke in Patients with Rheumatoid Arthritis 

Dear Dr. Lin:

I am pleased to inform you that your manuscript has been deemed suitable for publication in PLOS ONE. Congratulations! Your manuscript is now with our production department. 

With kind regards,

on behalf of

Dr. Seana Gall 

Academic Editor

PLOS ONE